# The Spatial Changes of Transportation Infrastructure and Its Threshold Effects on Urban Land Use Efficiency: Evidence from China

Xinhai Lu [1,2], Mengcheng Wang [1,*] and Yifeng Tang [1]

1 College of Public Administration, Huazhong University of Science and Technology, Wuhan 430074, China; luxinhai@hust.edu.cn (X.L.); d201881103@hust.edu.cn (Y.T.)
2 College of Public Administration, Central China Normal University, Wuhan 430079, China
* Correspondence: wmc@hust.edu.cn

**Abstract:** The development of social production and the agglomeration of the urban population have brought tremendous pressure to transportation infrastructure. However, the impacts of transportation development on urban land use systems have not been well investigated. Under the pressure of limited land resources, the impact of transportation infrastructure on urban land use efficiency (ULUE) is receiving increasing attention from scholars and needs to be explored. By collecting panel data from 30 regions in China from 2003 to 2018, in this study we constructed a spatial Durbin model and a panel threshold regression model to explore the spatial spillover effects and threshold effects of transportation infrastructure on ULUE. The most obvious findings emerging from this study are that (1) ULUE is not randomly distributed over different regions in China, but has an obvious positive spatial correlation; (2) transportation infrastructure has significant positive direct and spatial spillover effects on ULUE and the direct effects of transportation infrastructure (0.823) are significantly stronger than the spatial spillover effects (0.263); (3) the impact of transportation infrastructure on ULUE has a significant double threshold effect, and the threshold values are 4.520 and 6.429 respectively, and with the improvement of transportation infrastructure, its marginal effects on ULUE show a downward trend. This paper provides theoretical support for policymakers to achieve cross-regional cooperation on land use and transportation infrastructure construction and inspirations for sustainable development.

**Keywords:** transportation infrastructure; urban land use efficiency; spatial spillover effect; spatial Durbin model (SDM); threshold effect





## 1. Introduction

The past few decades have witnessed the fast and furious development of transportation infrastructure in China [1]. By redistributing accessibility in space, the fast and furious development of transportation infrastructure not only plays a critical role in land use transitions [2] but also accelerates the flow of socioeconomic factors [3,4], including population, technology, and information. However, in recent years, with the improvements in transportation, a trend of the fragmentation of land use patterns has emerged [5], and urban land use efficiency (ULUE) has also shown a downward trend [6]. Therefore, the question of how to achieve coordinated development between transportation infrastructure and ULUE to promote sustainable economic development remains unanswered and has become the focus of the attention of scholars in different disciplines.

Nowadays, there has been an increasing amount of research on transportation infrastructure. Several studies focus on the impact of transportation infrastructure construction on the process of urbanization [7], economic development [8], and industrial structure transformation [9]. Moreover, the relationship between transportation and ULUE has been explained. For example, macroeconomic production function models, such as the

Cobb–Douglas production function [10,11], are used to explore the impact of transportation infrastructure on land use. Moreover, some of the literature have evaluated railways [12], highways [13], high-speed railways [1,14], subways [15,16], and other transportation infrastructures' impacts on ULUE. However, the effect of transportation infrastructure at the provincial level is generally lower than the overall level of the country, which has triggered discussions about the spatial spillover effects of transportation infrastructure [17,18].

Although studies have directly or indirectly proved the effects of transportation infrastructures on land use, few studies have analyzed the spatial spillover effects on ULUE in detail. With the development of the spatial econometric panel data model, Anselin [19] proposed the spatial autoregression model (SAR) and the spatial error model (SEM) and developed the Lagrange Multiplier (LM) statistic to test the autocorrelation of the spatial lag term and the spatial error term. Zhang [20] adopted the SEM model to explore the impacts of entity and location on commercial real estate prices and achieved high precision and reduced the cost of the valuation. Hawkins and Habib [21] found that travel distance has great effects on land use patterns usingthe SAR model, and Wang et al. [22] used a spatial Durbin model to investigate the influences of both local and civil environmental regulation and its spatial spillover effects on green total factor productivity in 273 cities of China from 2003–2013. The spatial spillover effects of transportation infrastructures have been proven to exist [23,24]; however, conclusions are inconsistent. Boarnet [23] found that highways have obvious negative spillover effects between regions that compete with each other, that is, the construction of infrastructure in this region will transfer the production activities to neighboring regions, thus producing a negative spillover effect. Cohen and Paul [25] found that the development of transportation and other infrastructure in a certain region can reduce the transportation cost of neighboring areas and produce positive spatial spillover effects. On the contrary, Arbués et al. [26] tested the existence of direct and spillover effects of road, railway, airport, and seaport infrastructure projects and found that road transport infrastructure has positive effects on the output of the local region and its neighboring provinces, whereas other modes of transportation infrastructure have no significant impacts on average. Hulten et al. [24] found that the impact of transportation infrastructure is determined by its economic development stage by comparing the United States, India, and Spain. All in all, previous studies have rarely considered the spatial spillover effects of transportation infrastructure on ULUE.

Moreover, with the unprecedented transportation infrastructure development and urbanization in recent years in China, many scholars have begun to focus on the impact of transportation infrastructure on ULUE based on a linear regression model, but as in the case of many socio-economic systems, there is likely a diminishing marginal effect and the results have some limitations on practicality. However, there is some literature [27,28] indicating that transportation infrastructure of different density levels has nonlinear effects on economic development. Yang [29] employed a spatial Durbin model to investigate the nonlinear effects of environmental regulation on eco-efficiency under the constraint of land use carbon emissions. Zhang [30] found the distance threshold within which metro stations influence development intensity and the synergy between the presence of metro stations and land availability. Luo [31] found that the cross-regional operation of agricultural machinery has a positive impact on agricultural growth, and there is a threshold effect based on highway infrastructure construction. However, the literature on the nonlinear connection between transportation infrastructure and ULUE is still scarce. To fill the research gap mentioned above, we devoted this study to exploring the spatial impacts of transportation infrastructure on ULUE from the perspective of the spatial spillover effect; then, the nonlinear effects of transportation infrastructure on ULUE are explored by constructing a panel threshold model and the level of transportation infrastructure is used as a threshold variable. This paper provides theoretical support for governments to achieve cross-regional cooperation on land use and transportation infrastructure construction and provides inspiration for sustainable development.

The rest of this paper is arranged as follows: Section 2 introduces the mechanism and proposes the research hypothesis. In Section 3, we focus on the description of the study area, indicator selection, data sources, and research methods. In Section 4, the spatial Durbin model and panel threshold regression models are constructed to explore the spatial spillover effects and threshold effects of transportation infrastructure on ULUE. Sections 5 and 6 present a discussion and conclusions. Finally, the policy implications are given in Section 6.

## 2. Mechanism Analysis and Research Hypothesis

### 2.1. Mechanism Analysis

The improvement of ULUE mainly depends on the input level of land use factors. Transportation infrastructure, the mediating variable that affects traffic accessibility and the development of the social economy, plays a critical role in reshaping the pattern of land use. Specifically, the impact of transportation infrastructure on ULUE can be summarized as direct effects, spatial spillover effects, and threshold effects. A diagram of the influence mechanisms is shown in Figure 1.

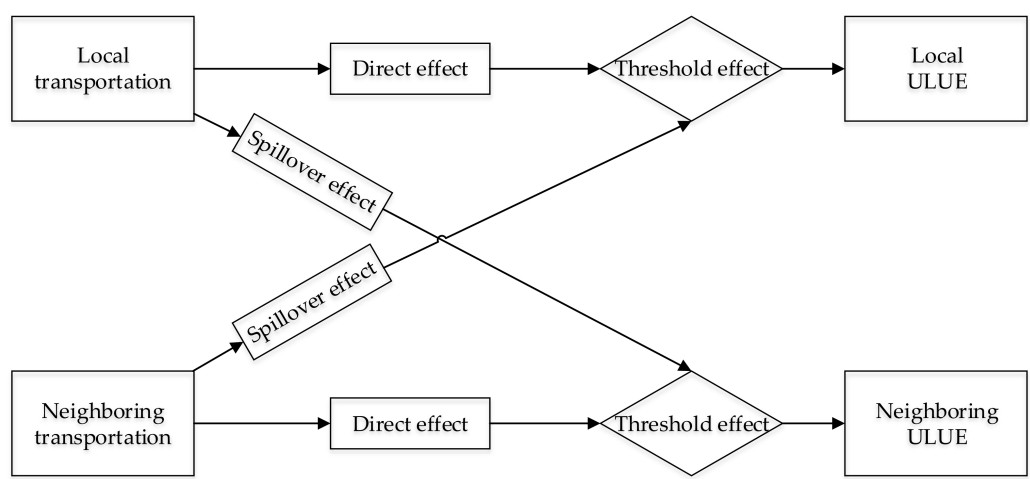

**Figure 1.** Diagram of the influencing mechanisms. ULUE, urban land use efficiency.

### 2.1.1. Direct Effect

Because of its accessibility, transportation infrastructure breaks the restrictions of spatial geography on land use and weakens the negative effects of geographic location [32]. The improvement of transportation infrastructure makes the exchange of socioeconomic factors required for land use more smooth and the space allocation more reasonable, and greatly reduces the cost of land use. Furthermore, transportation infrastructure can guide the distribution of land resources and other factors to high-yield industries, so that the industry structure can be further optimized, which directly promotes the improvement of ULUE in the local region. In this study, the above-mentioned effects were defined as the direct effects of transportation infrastructure on ULUE.

### 2.1.2. Spatial Spillover Effect

According to the first law of geography, proposed by Tobler [33], all things are related to others, but near things are more related to each other. Transportation infrastructure can break down the barriers of geography between regions and provide the opportunity for cross-regional cooperation in land use and social production by accelerating the flow of production factors such as capital investment and labor between regions and reducing inter-provincial transportation costs. In this way, land use in neighboring regions has gradually created spatial connections through cooperation and imitation. Notably, transportation infrastructure in the local region has indirect effects on the ULUE of its neighboring regions,

and its neighboring regions, at the same time, have indirect spillover effects on the ULUE of the local region. In this study, the indirect effects were defined as the spillover effects of transportation infrastructure on ULUE.

### 2.1.3. Threshold Effect

The impact of transportation infrastructure on production activities depends on its stock [34]. When it is below the threshold, there is almost no impact, and when it reaches a certain critical value in the current period, the impact may be very large. However, the effect of transportation infrastructure on economic growth is not endless. When the stock of transportation infrastructure reaches a certain limit, more construction will significantly reduce the role of transportation infrastructure in promoting economic development and ULUE. In this study, the above-mentioned effects were defined as the threshold effects of transportation infrastructure on ULUE.

### 2.2. Research Hypothesis

The improvement of transportation infrastructure is a long-term process of structural evolution. Changes in the stock of transportation infrastructure change the material circulation path and energy exchange mechanism of the urban land use system, which are directly reflected in the impact on the structure and function of urban land use. Using the law of diminishing returns to land [35], we can find the changes of ULUE under different levels of transportation infrastructure—when the transportation infrastructure is at a lower level, the flow of capital elements is limited, and the spatial allocation of labor elements is relatively insufficient. To meet the needs of urban development and social life, local governments tend to build more transportation infrastructure to make up for the disadvantages of land locations and achieve coordinated regional development. However, the improvement of transportation infrastructure is often accompanied by land use fragmentation [5] and unreasonable land use structure. Although ULUE is increasing, it is not at a high level. With the continuous development of the economy, the improvement of ULUE reduces the dependence on transportation infrastructure and the marginal impact of transportation infrastructure on ULUE gradually decreases. When the level of transportation infrastructure reaches a certain level, the marginal effects will continue to decline and ULUE will reach the highest point when the marginal effects approach zero. If it is further increased, ULUE may even decline.

Thus, this study puts forward the following hypothesis: With the development of transportation infrastructure, ULUE may show rapid growth first, then a slow growth, and then a trend of flattening or even declining.

## 3. Materials and Methods

### 3.1. Study Area

In the early stage of reform and opening up, China, as a developing country, had a lower level of transportation infrastructure, which severely hampered economic growth. In the early 1980s, China proposed the development of transportation infrastructure as the top priority of the national economy, which led to the rapid development of transportation infrastructure in China. There are 34 provinces in China, including Taiwan and two special administrative regions (Hong Kong and Macau) and five autonomous regions (Xinjiang, Ningxia, Tibet, Guangxi, and Inner Mongolia). Due to the limitations of data acquisition in Tibet, Taiwan, Hong Kong, and Macau, this study selects the remaining 30 provinces (regions) of China as the study area (Figure 2) and constructs a spatial econometric model and panel threshold regression model to explore the spatial impacts of transportation infrastructure on ULUE in China during the period 2003–2018.

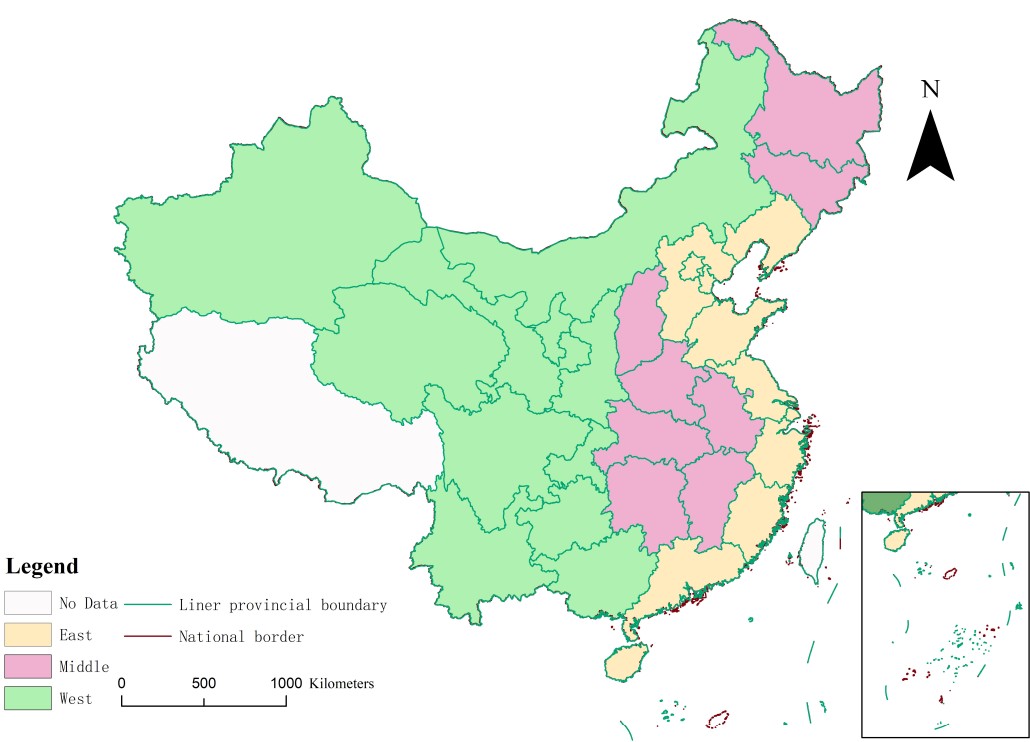

**Figure 2.** Study area.

*3.2. Indicators Selection*

3.2.1. Explained Variable: ULUE

ULUE refers to the added value of secondary and tertiary industries per unit of land area [36]. In this study, we selected the ratio of output values of the secondary and tertiary industries to each provincial land area as the index of ULUE. To eliminate the influence of price factors, the GDP of each province has been adjusted in this study. The temporal and spatial changes in ULUE is shown as Figure 3. The ULUE calculating formula is as follows:

$$ULUE_{it} = \frac{AVST_{it}}{Area_i} \qquad (1)$$

in Equation (1), $ULUE_{it}$ represents urban land use efficiency; $AVST_{it}$ represents the added value of the secondary and tertiary industries; and $Area_i$ represents the area of each province.

3.2.2. Explanatory Variable: Transportation Infrastructure

The transportation infrastructure in China mainly includes railways, highways, air transportation, water transportation, etc. However, railway and highway transportation bear more than 85% of China's freight and passenger volume, and the contribution of air and water transportation is relatively small; besides, the freight and passenger volume data for the inland waterways are missing. Considering these limitations on data acquisition, this study takes railway and highway transportation to represent transportation infrastructure. It is worth noting that railway transportation includes high-speed railways, fast railways, and ordinary railways, and highway transportation includes classified highways and substandard roads. Therefore, in this study, we take the sum of the density of railways and highways as the indicator used to represent the level of transportation infrastructure in China. The transportation calculating formula is as follows [37,38]:

$$Trans_i = \frac{Railway_i + Highway_i}{Area_i} \qquad (2)$$

in Equation (2), $Trans_i$ represents the level of transportation infrastructure of each province of China; *Railway* and *Highway* represent the mileage of railways and highways in each province respectively; and $Area_i$ represents the area of each province.

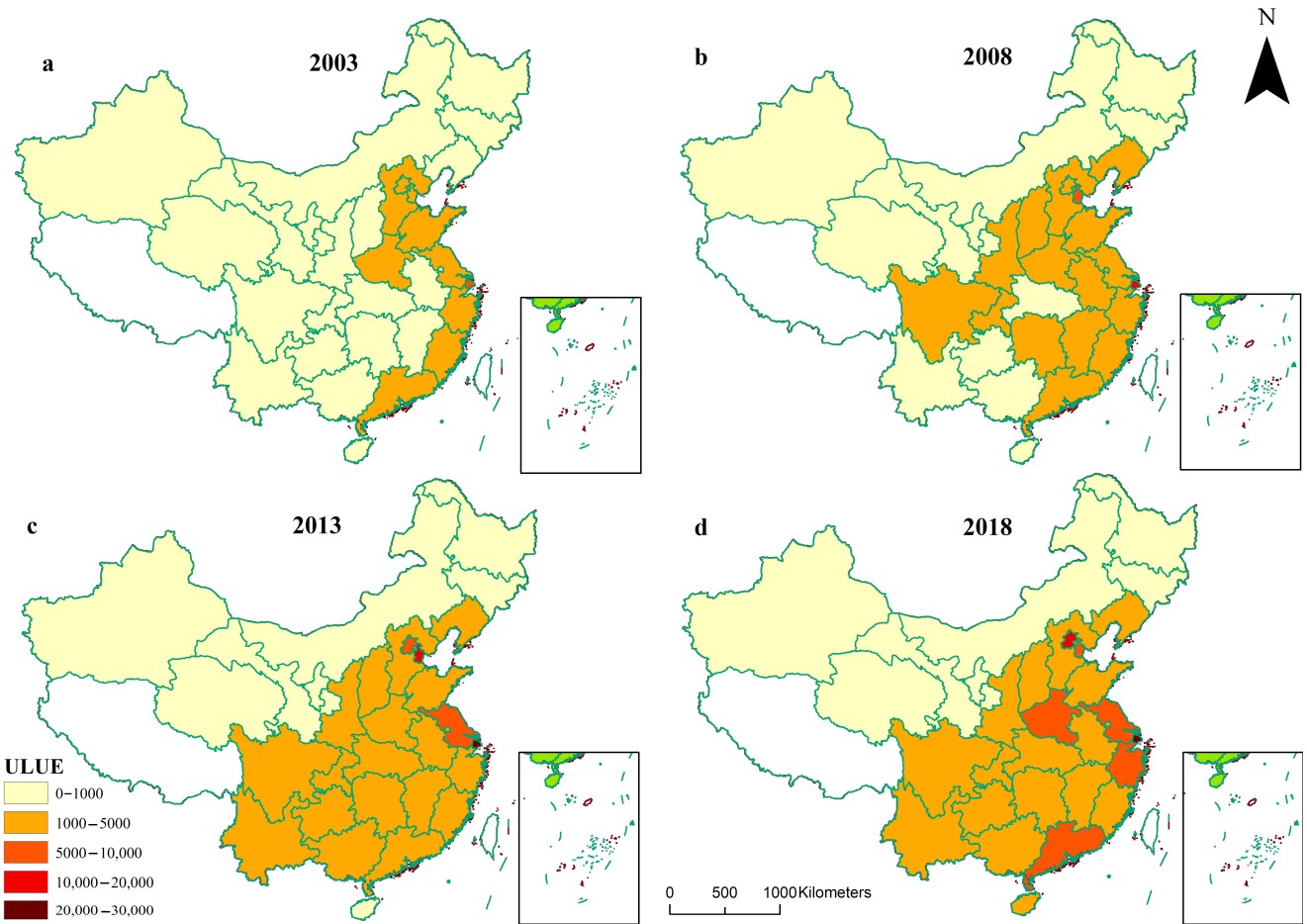

**Figure 3.** Temporal and spatial changes in ULUE. (**a**) ULUE in 2003; (**b**) ULUE in 2008; (**c**) ULUE in 2013; (**d**) ULUE in 2018.

### 3.2.3. Other Control Variables

ULUE is affected by not only transportation infrastructure but also many other factors. Referring to the existing studies [14,39], this study controlled the following variables:

First, labor input. The process of land use involves high labor intensity. In this study, the employed population of secondary and tertiary industries in each province represents the labor input, denoted as *Labor*.

Second, capital investment. The spatial allocation of capital investment will affect the land use mode and structure. This study uses the urban fixed assets investment of each provincial administrative region to represent the capital input, recorded as *Invest*.

Third, the industrial structure. The spatial difference of industrial structure can lead to different ULUE. In this study, we used the ratio of the total output value of second and third industries to the total GDP to represent the level of industrial structure, marked as *IS*.

### 3.3. Data Sources

The indicator data required for analysis were all derived from the China Statistical Yearbook, China Transportation Statistical Yearbook, China Urban Construction Statistical Yearbook, and provincial statistical yearbooks from 2004 to 2019. To reduce the dimensional influence and eliminate the possible heteroscedasticity between the indicators, we took the natural logarithm for these variables in the model. Using Stata 15.1 software, we calculated the mean, standard deviation, minimum, and maximum of all the variables, and the descriptive statistical results of each variable are shown in Table 1.

**Table 1.** Descriptive statistical results of the variables.

| Type | Variables | Symbol | Indicators | Mean | Standard Deviation | Minimum | Maximum |
|---|---|---|---|---|---|---|---|
| Explained variable | Urban land use efficiency | *ULUE* | The ratio of the added value of second and tertiary industries to each region's land area | 2673.611 | 3824.996 | 23.408 | 29,157.680 |
| Explanatory variable | Transportation infrastructure | *Trans* | Weighted railway and highway density | 2.525 | 1.632 | 0.170 | 7.440 |
| Control variable | Labor input | *Labor* | The employed population of secondary and tertiary industries | 1601.453 | 1175.240 | 133.020 | 5159.720 |
| | Capital investment | *Invest* | Urban fixed asset investment | 10,305.210 | 10,796.110 | 236.900 | 56,459.710 |
| | Industry structure | *IS* | The ratio of the added value of second and tertiary industries to the total GDP | 0.425 | 0.090 | 0.290 | 0.810 |

### 3.4. Research Method

#### 3.4.1. Spatial Autocorrelation Test

The above analysis indicates that ULUE in the local area may be affected by transportation of its neighboring areas, that is, there may be a spatial correlation. In econometrics, the spatial correlation is often measured using the global Moran's index (Moran's I). The Global Moran's index reflects the degree of similarity of the attribute values among spatially adjacent regions and its value range is (−1,1). Moran's I > 0 indicates a positive spatial correlation, Moran's I < 0 indicates a negative spatial correlation, and when Moran's I = 0, it means there is no spatial correlation. The calculation formula is as follows:

$$Moran's\ I = \frac{\sum\limits_{i=1}^{n}\sum\limits_{j\neq i}^{n} W_{ij}\left(d_i - \bar{d}\right)\left(d_j - \bar{d}\right)}{S^2 \sum\limits_{i=1}^{n}\sum\limits_{j\neq i}^{n} W_{ij}}, S^2 = \frac{\sum\limits_{i=1}^{n}\left(d_i - \bar{d}\right)^2}{n} \tag{3}$$

in Equation (3), $d_i$ and $d_j$ represent the ULUE of region $i$ and region $j$, respectively; $n$ is the number of regions; $W_{ij}$ is the spatial weight matrix of N×N; $\bar{d}$ is the mean value of the samples; and $S^2$ is the variance of the samples.

In the spatial econometric analysis, it is necessary to introduce spatial weighting matrices to describe the relationships among different regions in China. To explore the spatial correlation, this study calculates the binary contiguity spatial weight matrix $W_{ij}$. The setting principle of $W_{ij}$ is:

$$W_{ij} = \begin{cases} 1 & \text{when region i is adjacent to region j} \\ 0 & \text{when region i is not adjacent to region j} \end{cases} \quad i, j = 1, 2, \cdots, n \tag{4}$$

#### 3.4.2. Spatial Econometric Model

This study uses spatial econometric models to analyze the impact of transportation infrastructure on ULUE in China and to measure the direct effects and spatial spillover effects. The spatial econometric model includes three forms—the spatial autoregression model (SAR), spatial error model (SEM), and spatial Durbin model (SDM). The SAR only includes the lag term of the spatial dependent variable, and the SEM only includes spatial spillover effects of independent variables, whereas the SDM includes both the lag term of the spatial dependent variable and the spatial spillover effects of independent variables. Based on this, these three spatial econometric models are constructed as follows [40]:

SAR:

$$\ln(ULUE_{it}) = \alpha + \rho \sum_{j=1}^{n} W_{ij} * \ln(ULUE_{it}) + \beta_i \ln(X_{it}) + \varepsilon_{it} \tag{5}$$

SEM:

$$\ln(ULUE_{it}) = \alpha + \beta_i \ln(X_{it}) + \lambda W_{ij}\mu_{it} + \varepsilon_{it} \tag{6}$$

SDM:

$$\ln(ULUE_{it}) = \alpha + \rho \sum_{j=1}^{n} W_{ij} * \ln(LUE_{it}) + \beta_i \ln(X_{it}) + \theta_i \sum_{j=1}^{n} W_{ij} * \ln(X_{it}) + \varepsilon_{it} \quad (7)$$

in Equations (5)–(7), $ULUE_{it}$ represents urban land use efficiency; $i$ and $j$ represent regions; $t$ represents years; $X_{it}$ represents the explanatory variable matrix, including transportation; $W_{ij}$ represents the N × N spatial weight matrix; $W_{ij}$ *ln($ULUE_{it}$) represents the spatial lag of land use efficiency; and $W_{ij}$ *ln(Xit) represents the spatial lag of explanatory variables. $\varepsilon_{it}$ and μit are normally distributed random error vectors; $\alpha$ denotes the intercepted item; $\rho$, $\beta$, and $\theta$ represent the regression coefficient; and $\lambda$ represents spatial error coefficient.

### 3.4.3. Model Selecting Tests

This study adopts the Lagrange multiplier (LM) test proposed by Anselin [19] to test whether the model we constructed contains spatial interaction. At the same time, due to the introduction of the spatial lag term in the spatial econometric model, it is necessary to consider the marginal influence of the explanatory variables when further interpreting the regression coefficient of the spatial econometric model. According to Elhorst [41], when $\rho \neq 0$ but $\theta = 0$, SDM can be simplified into SAR; when $\lambda = 0$ and $\rho = 0$, SDM can be simplified into SEM; when $\rho \neq 0$ and $\theta \neq 0$, but $\lambda = 0$, SDM should be adopted.

However, the coefficient $\theta$ of the explained variable matrix cannot fully measure the influence of the explanatory variable on the explained variable. It is necessary to decompose the total effects of the spatial model into direct effects and spatial spillover effects by means of the partial differentiation method [40]. The formulas are as follows:

$$\overline{M}(r)_{total} = n^{-1} I_n S_r(W)_{I_n} \quad (8)$$

$$\overline{M}(r)_{direct} = n^{-1} \sum_{n=1}^{n} \sum_{n=1}^{n} S_r(W)_{nn} \quad (9)$$

$$\overline{M}(r)_{spillover} = \overline{M}(r)_{total} - \overline{M}(r)_{direct} \quad (10)$$

In Equations (8)−(10), $\overline{M}(r)_{total}$, $\overline{M}(r)_{direct}$, and $\overline{M}(r)_{spillover}$ represent the total effects, direct effects, and spillover effects of transportation infrastructure on urban land use efficiency; $i$ and $j$ represent regions; n represents the number of regions; $W_{ij}$ represents the N×N spatial weight matrix; and $S_r(W_{ij})$ represents the impacts of the explanatory variable $x_{rj}$ on the explained variable in region $i$.

### 3.4.4. Panel Threshold Regression Model

The existing research shows that there may be a non-linear relationship between transportation infrastructure and economic development, so in this study, a panel threshold regression model was adopted to test whether the threshold effects of transportation infrastructure on ULUE exist. Considering that the transportation infrastructure of the local region will have spatial spillover effects on ULUE of its neighboring regions, and the transportation infrastructure of its neighboring regions will also have spatial spillover effects on ULUE of the local region, in this study we took the sum of the total density of railways and highways in local regions (*Trans_local*) and the spatial spillover effects of the transportation infrastructure of other regions on local ULUE ($W_{ij}$* *Trans_neighbor*), which is calculated in the spatial Durbin model, as the threshold explanatory variable for threshold effect regression, denoted as *TotalTrans*. *TotalTrans* is also the core explanatory variable in this panel threshold regression model. The calculating formula of *TotalTrans* and the panel threshold regression model [42] is as follows:

$$TotalTrans = Trans_{local} + W_{ij} * Trans_{neighbor} \quad (11)$$

$$\begin{aligned} \ln(ULUE_{it}) &= \mu_i + \alpha_1 TotalTrans_{it} * I(TotalTrans_{it} < \gamma_1) \\ &+ \alpha_2 TotalTrans_{it} * I(\gamma_1 \leq TotalTrans_{it} < \gamma_2) \\ &+ \alpha_3 TotalTrans_{it} * I(TotalTrans_{it} \geq \gamma_2) + \lambda X_{it} + e_{it} \end{aligned} \quad (12)$$

In Equations (11) and (12), *TotalTrans* represents the level of transportation infrastructure and $ULUE_{it}$ represents urban land use efficiency. $I(\cdot)$ is an indicative function;

$\gamma_1$, $\gamma_2$ are the thresholds to be estimated, $\alpha_1$, $\alpha_2$, and $\alpha_3$ are the influence coefficients of TotalTrans; $X_{it}$ is the control variable matrix; $\lambda$ is the influence coefficient of $X_{it}$; and $e_{it}$ is the random disturbance term. To reduce multicollinearity and heteroscedasticity, this study takes the natural logarithm of related variables. Due to space limitations, only the double-threshold regression model is listed here, but in the part of the empirical analysis below, all the threshold values are tested and reported in this paper.

## 4. Results and Discussion

### 4.1. Results of Spatial Autocorrelation Test

To verify whether there is a spatial correlation of ULUE among different provinces, the global Moran's index was used in this study. Table 2 reports the changes in global Moran's index of ULUE. The results showed that there was a significant positive spatial correlation in all years, which is consistent with the results of the existing research. Thus, it can be preliminarily determined that ULUE has spatial autocorrelation from a global perspective, which is one of the prerequisites for using a spatial econometric model to explore the spatial spillover effects of ULUE [40].

**Table 2.** Results of ULUE spatial autocorrelation from 2003–2018. Moran's I, global Moran's index.

| Year | Moran's *I* | Z-Value | *p*-Value | Year | Moran's *I* | Z-Value | *p*-Value |
|------|-----------|---------|-----------|------|-----------|---------|-----------|
| 2003 | 0.166 | 2.822 | 0.0048 | 2011 | 0.195 | 2.715 | 0.0066 |
| 2004 | 0.171 | 2.873 | 0.0041 | 2012 | 0.203 | 2.720 | 0.0065 |
| 2005 | 0.172 | 2.759 | 0.0058 | 2013 | 0.201 | 2.672 | 0.0075 |
| 2006 | 0.180 | 2.790 | 0.0052 | 2014 | 0.198 | 2.649 | 0.0081 |
| 2007 | 0.180 | 2.864 | 0.0042 | 2015 | 0.194 | 2.835 | 0.0046 |
| 2008 | 0.124 | 3.207 | 0.0013 | 2016 | 0.189 | 2.906 | 0.0037 |
| 2009 | 0.181 | 2.723 | 0.0065 | 2017 | 0.192 | 2.950 | 0.0032 |
| 2010 | 0.187 | 2.713 | 0.0067 | 2018 | 0.190 | 2.940 | 0.0033 |

### 4.2. Results of Model Selection Tests

Through the above analysis, it was determined that ULUE does have a spatial correlation between provinces; thus, we selected the suitable model using LM, Wald, likelihood ratio (LR), and Hausman tests. Table 3 reports the results of the LM test and the robust LM test under four kinds of conditions: no fixed effects, space-fixed effects, time-fixed effects, and space and time double-fixed effects. The results showed that the spatial econometric model that is based on spatial fixed effects is significant at the 5% and 1% significance levels, respectively. Meanwhile, The Wald and LR test results (Table 4) showed that the spatial Durbin model cannot be simplified into SAR or SEM at the 1% significance level. At the same time, the Hausman test results (Table 4) showed that the null hypothesis that the model has random effects can be rejected at the 1% significance level, indicating that the spatial Durbin model with spatial fixed effects should be adopted in this study [41].

**Table 3.** Results of the Lagrange multiplier (LM) test and robust LM test.

| Test | No Fixed Effects | Spatial Fixed Effects | Time Fixed Effects | Spatial and Time Double Fixed Effects |
|------|------------------|-----------------------|--------------------|---------------------------------------|
| LM spatial lag | 83.950 *** | 3.953 ** | 49.773 *** | 0.720 |
| Robust LM spatial lag | 3.053 ** | 6.212 ** | 39.162 *** | 30.817 *** |
| LM spatial error | 154.743 *** | 31.951 *** | 10.719 *** | 7.324 *** |
| Robust LM spatial error | 73.845 *** | 34.209 *** | 0.108 | 37.420 *** |

Note: * represents $\rho < 10\%$, ** represents $\rho < 5\%$, and *** represents $\rho < 1\%$, respectively.

**Table 4.** Results of the Wald, likelihood ratio (LR), and Hausman tests.

| Test | Wald Test-Spatial Lag | LR Test-Spatial Lag | Wald Test-Spatial Error | LR Test-Spatial Error | Hausman |
|------|-----------------------|---------------------|-------------------------|-----------------------|---------|
| Result | 105.009 *** | 90.404 *** | 86.518 *** | 80.419 *** | 79.790 *** |

Note: * represents $\rho < 10\%$, ** represents $\rho < 5\%$, and *** represents $\rho < 1\%$, respectively.

### 4.3. Results of Direct and Spatial Spillover Effects

Following the method of LeSage and Pace [40], we decomposed the results of the spatial Durbin model into direct, spatial spillover, and total effects using Matlab R2014b software [43]. The results are shown in Table 5. The results indicated that transportation infrastructure has significant positive direct and spatial spillover effects on ULUE. The coefficient of the direct effects of transportation on the ULUE was 0.823, accounting for 75.7% of the total effects, and the coefficient of the spatial spillover effects of transportation on the ULUE was 0.264, accounting for 24.3% of the total effects. That is to say, when the level of transportation infrastructure increases by 1%, ULUE in the local province will increase by 0.823% and that of the neighboring areas will increase by 0.264%.

**Table 5.** Estimation results of the spatial Durbin model.

| Variables | Direct Effects | Spatial Spillover Effects | Total Effects |
|---|---|---|---|
| ln(*Trans*) | 0.823 *** (20.42) | 0.264 *** (7.36) | 1.087 *** (22.96) |
| ln(*Labor*) | 0.057 *** (1.00) | 0.018 (0.99) | 0.075 (1.00) |
| ln(*Invest*) | 0.164 ** (2.50) | 0.053 ** (2.21) | 0.217 ** (2.45) |
| ln(*IS*) | 0.402 * (2.00) | 0.130 * (1.83) | 0.532 * (1.98) |

Note: * represents $\rho < 10\%$, ** represents $\rho < 5\%$, and *** represents $\rho < 1\%$, respectively. Value in parentheses is the *t*-test value.

The results also demonstrated that ULUE has a strong dependence on the transportation infrastructure, which is consistent with the findings of Xie and Wang [44]. The improvement of transportation infrastructure will improve the accessibility of land in the region, and reduce the transportation and transaction costs of production factors in the process of land use, and ultimately promote the ULUE of the local region. Moreover, the transportation infrastructure connects the economics of various regions as a whole, and through the diffusions and exchanges of production factors between regions [3], factors of production in more developed regions flow to the less developed neighboring regions. One developed region can drive the development of its adjacent regions, which is the reason why transportation shows positive spatial spillover effects on the ULUE of its neighboring areas. Further comparison showed that the direct effects of transportation infrastructure (0.823) are significantly stronger than the spatial spillover effects (0.263), nearly three times the spatial spillover effects. This is because the overall level of China's transportation infrastructure is still low at present [28] and the direct effects of transportation infrastructure on local ULUE still dominate the total effects.

As for the control variables, the labor input has positive direct effects on ULUE, whereas its spatial spillover effects are positive but not significant, which is consistent with Tang [45]. On the one hand, current land use methods in China are always rough and labor-intensive and the present labor input has exceeded the land-carrying capacity, and the excessive labor input can result in a decline in ULUE in the local and neighboring provinces. On the other hand, various and complex human resource policies in different regions can be confusing for less educated laborers, and the increasing labor provincial migration cost can greatly reduce the enthusiasm for labor migration.

The direct and spatial spillover effects of capital input and industrial structure were all positive and significant. These developed provinces tend to achieve agglomeration economic effects by connecting themselves and their neighbors into a whole and optimizing the spatial allocation of capital investment and industrial structure in a larger scope, which allows capital, technology, and other product factors to diffuse to the neighboring areas [3], making the ULUE of the whole area higher.

### 4.4. Results of the Panel Threshold Regression Model

Before constructing the panel threshold regression model, we should determine whether the threshold effect exists. If it exists, then the threshold values can be further estimated [42]. Using Stata 15.1 software, the thresholds were tested 1000 times by bootstrapping, and the results are shown in Table 6.

**Table 6.** The results of the threshold effect tests.

| Number of Thresholds | F-Statistics | Bootstrap | Critical Value | | | Threshold Value | 95% Confidence Interval |
|---|---|---|---|---|---|---|---|
| | | | 10% | 5% | 1% | | |
| Single | 97.41 *** | 1000 | 38.214 | 45.188 | 64.072 | 5.867 *** | (5.763, 5.877) |
| Double | 42.71 ** | 1000 | 36.812 | 46.639 | 65.501 | 4.520 ** | (4.454, 4.549) |
| | | | | | | 6.429 ** | (5.999, 6.496) |
| Triple | 16.48 | 1000 | 47.276 | 56.121 | 79.983 | | |

Note: * represents $\rho < 10\%$, ** represents $\rho < 5\%$, and *** represents $\rho < 1\%$, respectively.

As shown in Table 6, the F-statistics of the single threshold regression model and the double threshold regression model were significant at the 1% and 5% level, respectively, whereas the F-statistics of the triple threshold regression model were not significant, which indicates that the impact of transportation infrastructure on ULUE has a significant double threshold effect, and the threshold values are 4.520 and 6.429, respectively.

At the same time, the corresponding threshold value maximum likelihood estimation test charts we drew using Stata 15.1 software are shown in Figure 4. The lowest value in the figure is the threshold value in the likelihood ratio (LR). The likelihood ratio of the threshold variable is represented by a solid line, and the threshold values at the significance level of 5% are represented by a dashed line, which also verifies the existence of a double threshold. Finally, the panel threshold regression estimation results are reported in Table 7.

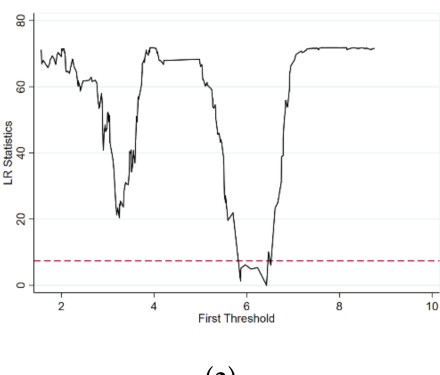
(**a**)

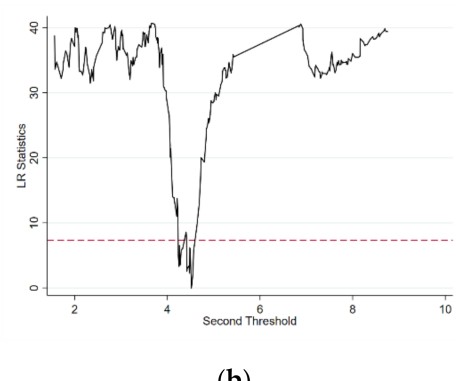
(**b**)

**Figure 4.** Results of double threshold LR estimation. (**a**) Results of the first threshold LR estimation; (**b**) Results of the second threshold LR estimation.

**Table 7.** Results of the double threshold regression model.

| Explanatory Variables | Coefficient | Standard Error | T-Statistics | 95% Confidence Interval |
|---|---|---|---|---|
| *TotalTrans* (*TotalTrans* < 4.520) | 0.192 *** | 0.013 | 14.37 | (0.166,0.218) |
| *TotalTrans* (4.520 ≤ *TotalTrans* < 6.429) | 0.149 *** | 0.010 | 14.99 | (0.129,0.168) |
| *TotalTrans* (*TotalTrans* ≥ 6.429) | 0.109 *** | 0.007 | 14.78 | (0.094, 0.123) |
| ln(*Labor*) | 0.216 *** | 0.055 | 3.95 | (0.109, 0.324) |
| ln(*Invest*) | 0.137 *** | 0.020 | 6.88 | (0.098, 0.176) |
| ln(*IS*) | −0.444 *** | 0.072 | −6.15 | (−0.586, −0.302) |
| Cons_ | −2.784 *** | 0.564 | −4.94 | (−3.892, −1.676) |
| $R^2$ | 0.948 | | | |

Note: * represents $\rho < 10\%$, ** represents $\rho < 5\%$, and *** represents $\rho < 1\%$, respectively.

Based on the threshold regression results (Table 7), it is obvious that transportation infrastructure does have a positive and significant effect on ULUE. To be specific, when *TotalTrans* < 4.520, the regression coefficient of transportation infrastructure is 0.192, which means the effects that transportation exerts on ULUE will increase by 0.192% for every 1% increase in transportation; when $4.520 \leq TotalTrans < 6.429$, these promoting effects are reduced to 77.6% of the first stage (0.149); and when $TotalTrans \geq 6.429$, its promoting effects are further reduced to 0.109, only 56.8% of the first stage. Based on the above analyses, this study divided 30 regions of China into three intervals, including regions with less-developed transportation infrastructure, regions with middle-developed transportation infrastructure, and regions with highly-developed transportation infrastructure. We took 2003 and 2018 as examples to reflect the features of the time evolution of transportation infrastructure (Table 8). Generally speaking, transportation infrastructure has positive effects on ULUE, and as the level of transportation infrastructure increases, the promoting effect shows a downward trend. This result is also in line with the law of diminishing marginal returns. In the year 2003, all 30 regions in China had less developed transportation infrastructures, and at that time, there was a large room for improvement of the level of transportation infrastructure to promote the provincial economic exchanges and the flow of factors of production between regions [28]. When China began to implement a proactive fiscal policy to support the construction of transportation infrastructure in various regions, the development of China's transportation infrastructure began to enter the fast lane. The marginal effects of transportation infrastructure on ULUE are the most obvious. That is why the development of transportation infrastructure by local governments plays a strong role in promoting ULUE. Furthermore, the promoting effects in regions with middle-developed and highly-developed transportation infrastructure are decreasing because the level of transportation infrastructure in various regions of China is continuously improving, making its marginal impact on ULUE decrease—this is because with the density of transportation infrastructure increasing, the land use pattern is becoming fragmented and the scale and agglomeration effects are gradually disappearing [5], resulting in a lower ULUE.

**Table 8.** Division of transportation infrastructure.

| Interval Division | 2003 | 2018 |
|---|---|---|
| Regions with less developed transportation infrastructure | Beijing, Tianjin, Hebei, Shanxi, Inner Mongolia, Liaoning, Jilin, Heilongjiang, Shanghai, Jiangsu, Zhejiang, Anhui, Fujian, Jiangxi, Shandong, Henan, Hubei, Hunan, Guangdong, Guangxi, Hainan, Chongqing, Sichuan, Guizhou, Yunnan, Shaanxi, Gansu, Qinghai, Ningxia, Xinjiang | Beijing, Tianjin, Inner Mongolia, Liaoning, Jilin, Heilongjiang, Shanghai, Hainan, Gansu, Qinghai, Ningxia, Xinjiang |
| Regions with moderately-developed transportation infrastructure | | Jiangsu, Zhejiang, Fujian, Guangdong, Gongxi |
| Regions with highly-developed transportation infrastructure | | Hebei, Shanxi, Anhui, Jiangxi, Henan, Shandong, Hubei, Hunan, Chongqing, Sichuan, Guizhou, Yunnan, Shaanxi |

### 4.5. Discussion

Existing research has proved that transportation infrastructure in the local region has a significant positive impact on the efficiency of ULUE [1,46] and may have an impact on its neighboring areas [38], based on linear models. In this study, we used spatial econometrics methods to explore the nonlinear effects of China's transportation infrastructure on ULUE. The test results show that transportation infrastructure has a significant double threshold effect on ULUE; that is, when the improvement of transportation infrastructure reaches a certain limit, it will cause the marginal impact of transportation infrastructure on ULUE to decrease gradually, which is consistent with the research hypothesis of this study.

However, there are still limitations of this study. Firstly, the mechanism of transportation infrastructure on urban land use and the potential intermediary effects are

not discussed in depth in this study, which is also one of the future research directions. Secondly, in this study, we have only explored the impacts of the total transportation infrastructure on ULUE, but have not discussed the individual impact of railway, highways, and other modes of transport on ULUE. Moreover, due to data limitations, this study only concentrates on impacts at the provincial level. The question of whether our findings are also applicable to the city or county level requires more research and methods to be fully explored and analyzed. Nonetheless, our research results still have a certain significance by providing inspiration for governments to reduce the cost of transportation infrastructure construction, adjust the modes of land use, and ensure that the regional transportation infrastructure can have a positive impact on ULUE.

## 5. Conclusions

This study used the spatial Durbin model and panel threshold regression model to analyze the panel data of 30 regions in China from 2003 to 2018, and examined the threshold effects of China's transportation infrastructure on ULUE through empirical methods. The main conclusions drawn were as follows:

(1) ULUE had a significant positive spatial correlation at the provincial level in China from 2003 to 2018; that is, the ULUE of each province was not randomly distributed in space, but was influenced by its neighboring regions.

(2) The construction of transportation infrastructure facilitates the agglomeration of population and industries and optimizes the spatial allocation of production factors. At the same time, the construction of transportation infrastructure also connects the regions as a whole and accelerates the cross-regional exchange of socioeconomic factors. That is to say, the construction of transportation infrastructure not only improves ULUE in the local region but also has positive spatial spillover effects on the growth of ULUE in its neighboring regions.

(3) Transportation infrastructure has a significant threshold effect on ULUE. When the level of transportation infrastructure reaches a certain level, the marginal effects of transportation infrastructure on ULUE continue to decline in stages.

## 6. Policy Implications

According to the above research conclusions, we have drawn some policy implications as follows:

(1) Transportation infrastructure has a significant spatial spillover effect on ULUE. The construction of transportation infrastructure has improved the urban traffic conditions, reduced the cost of cross-region travel time, accelerated the flow of socio-economic factors, and improved the agglomeration of population and industry, and ultimately improved ULUE. Governments at all levels should break the administrative monopolies and achieve coordinated regional development in the field of the construction of transportation infrastructure.

(2) Due to the threshold effects of transportation infrastructure on ULUE, the central government is required to implement differentiated transportation infrastructure investment strategies based on the socio-economic development conditions of different regions. For regions with less-developed transportation infrastructure, such as Gansu, Qinghai, Ningxia, and Xinjiang, policy support should be given to strengthening the construction of transportation infrastructure to eliminate the bottleneck restriction of transportation infrastructure on ULUE and strengthen transportation connectivity with the eastern regions with moderately-developed and highly-developed transportation infrastructure to promote the rational allocation of production factors for urban land utilization.

(3) Policymakers in regions with moderately-developed and highly-developed transportation infrastructure, such as Jiangsu, Zhejiang, Hubei, and Hunan, are required to master the balance between the stock and flow of transportation infrastructure in the planning process, seek a reasonable spatial layout of transportation infrastructure among regions, and take effective measures to control the risk of disorderly construction of trans-

portation infrastructure to avoid resource wastage caused by the pursuit of excessively high levels of transportation infrastructure, which coincides with the United Nations Sustainable Development Goals. Although state land ownership in China is significantly different from private land ownership in most countries, the abovementioned impacts of transportation infrastructure on ULUE are applicable to different land ownership systems [12,25].

**Author Contributions:** Conceptualization, X.L., M.W. and Y.T.; data curation, X.L. and M.W.; formal analysis, X.L. and M.W.; funding acquisition, X.L.; methodology, X.L., M.W. and Y.T.; project administration, X.L., M.W. and Y.T.; software, M.W.; supervision, X.L.; visualization, X.L. and M.W.; writing—original draft, X.L., M.W. and Y.T.; writing—review and editing, M.W. All authors have read and agreed to the published version of the manuscript.

**Funding:** This research was supported by the National Natural Science Foundation of China (Nos. 71673096, 41801205), the China Postdoctoral Science Foundation Project (Nos.2020M672365), the National 985 Project of Nontraditional Security at Huazhong University of Science and Technology, and the Graduates' Innovation Fund, Huazhong University of Science and Technology (2020yjsCXCY069).

**Institutional Review Board Statement:** Not applicable.

**Informed Consent Statement:** Not applicable.

**Data Availability Statement:** The data presented in this study are available on request from the corresponding author. The data are not publicly available due to privacy.

**Acknowledgments:** The authors would like to thank the reviewers and the editor whose suggestions greatly improved the manuscript.

**Conflicts of Interest:** The authors declare no conflict of interest.

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
