# Peer review of "The Spatial Changes of Transportation Infrastructure and Its Threshold Effects on Urban Land Use Efficiency: Evidence from China"

_land, doi:10.3390/land10040346_

Round 1
Reviewer 1 Report
Dear Authors, the manuscript is well structure. It gets relevance by aiming to fill a considerable research gap, as most of the existing studies in China “have explored the impact of transportation infrastructure on ULUE based on a linear regression model”, but revealing results that presented have some practical limitations. Cumulatively, literature focusing on the nonlinear connection between transportation infrastructure and ULUE is still scarce (as mentioned in lines 65-66).
However, some suggestions/improvements can be done.
In lines 349-351, authors wrote: “Generally speaking, transportation infrastructure has positive effects on ULUE, and as the level of transportation infrastructure increases, the promoting effect shows a downward trend”. Due to its centrality and importance, this statement should be better explained/ justified.
Regarding the theme of this study, which has a strong relationship with environmental sustainability, this article would benefit from an authors’ note (probably in the “6. Policy implications” section) on how this kind of studies frame the UN Sustainable Development Goals (SDGs) indicator 11.3.1 which is designed to test urban land use efficiency. Also,
Discussion section should be enlarged. Even considering the limitations of the study (“However, there are still limitations of this study. The index of transportation infrastructure may be relatively simple and needs to be deepened”), it is a fact that authors referrer in the end of the Introduction section that “This paper provides theoretical support for governments to achieve cross-regional cooperation on land use and transportation infrastructure construction.” Saying that, other topics should be clear stated as not being possible to achieve in the study results (e.g.: no reference has been made regarding the need for more study/research on the indirect Impacts of Transportation Infrastructure on ULU; OR, does this study can help transportation planning decisions to determine which congestion reduction measures can support province strategic objectives, and therefore, help to reduce infrastructure costs, adjust land use patterns and assure positive effects on ULUE at local or regional scale.
Reviewer 2 Report
The current study is an important one for Urban Land Use Efficiency. Increasing population is also inducing pressure of transportation. Many socioeconomic factors also become accelerated due to fast transportation. Although the study is well conducted, there are some major deficiencies. I suggest a major revision for this article (land-1150874-peer-review-v1).
1- The title is not suitable for the manuscript. It is better to use spatial changes. Please specify that what was the purpose of study?
2- Systematic abstract is missing. An abstract must contain an introduction, identified problem with the study's aim, quantitative results, and conclusion with future recommendations. No statement of the problem is present in the abstract. Please give a clear-cut problem statement before going to the aim of the study in the abstract. Also provide quantitative results data.
3- I have doubts about hypothesis. Please specify that.
4- No statistical analysis details are provided in the manuscript. Please specify also if any model is use for assessment.
5- The discussion part has low and unnecessary information. Authors must focus on those attributes they have studied and the factors identification behind them—no need to explain what other scientists have observed.
6- No conclusive conclusion and potential benefits of this study are provided—also, elaborate future protective of this study. Conclusively relate the aims with the conclusion.
Minor
The English language needs extensive editing.
Reviewer 3 Report
General remarks:
- I have read the paper thoroughly. It focuses on an important topic and uses quantitative methods.
- The paper lacks a precise research question or hypothesis. Authors' statement "To fill the research gap mentioned above, we devoted this study to explore the impact of transportation infrastructure on ULUE from the perspective of the spatial spillover effect; then the nonlinear effects of transportation infrastructure on ULUE are explored by constructing a panel threshold model, and the level of transportation infrastructure is used ad threshold variable" [rows 66-71] is not enough in this case.
- Methods are "lightly" mentioned in the introductory part should be discussed, and the selection of a spatial Durbin model should be judged [rows 50-51].
- The paper lacks a literature review section, although relations between transport infrastructure and land use are well deployed in the scientific discussion over the years. Moreover, according to China, the phenomena of the rapid development of the high-speed railway and its impact on a land-use is well researched in the scientific literature. I recommend to use more literature in the Discussion section instead of developing a separate section on the literature review. In general, I find a relatively limited amount of the literature as a substantial shortcoming of the paper.
- Lack of input data – Authors should provide input data for the 30 regions as an attachment that could be easily accessible by other researchers to verify results.
- How have the Authors merged data for railway and road transport infrastructure? In the case of railway, have they considered the whole rail network (high-speed railway & "conventional" lines) into account? The method lacks a more detailed description. Authors focused on presenting formulas and the scientific process, whereas data input and its description were substantially underestimated. It is the substantial shortcoming of the paper.
- Discussion is minimal and does not show the desired scientific quality. Authors limit Discussion to general remarks without in-depth discussion with other authors. Lack of literature review is even more visible when the discussion part is so limited. This part must be improved, as it is another essential shortcoming of the paper.
- Conclusions are general, and they do not present differences between regions. Are there several similar "clusters"/groups of the regions? Are all regions similar in the case of the results obtained? Author lack with the presentation of the results in a more precise/detailed way.
- The whole part presenting the study's limitations should be removed from the discussion to separate chapter or integrated with conclusions as a separate subchapter.
- Conclusion nr 3 [rows 377-379] "(3) The impact of transportation infrastructure on ULUE has a significant double threshold effect. When the level of transportation infrastructure reaches a certain limit, the promotion of transportation infrastructure on ULUE will gradually decrease" seems to be unclear. Was a "promotion" a factor already included into analysis? What is the definition of "promotion" in this specific case?
Formal remarks:
- The numbering of parts is incorrect and inconsistent
- In the equation (2) there is an expression "hailway" which should probably be "highway"?
Reviewer 4 Report
There is a strong interaction between the transport system and the land use planning of the area. Therefore, the sustainable economic development of the region requires the coordination of activities related to the development of transport infrastructure and urban land use efficiency (ULUE). The issue is very important and has been undertaken as a research problem for many decades by scientists from around the world. However, due to the diversity of the obtained results, it requires further research and application of other approaches. This justifies taking up the subject by the Authors.
The manuscript is written in logical manner and the conclusions are justified. The Authors cited 39 literature sources, the vast majority of which are recent publications.
The specific comments are listed below:
- The introduction should be supplemented with the simple overwiew and the short description of the content of each section.
- The numbering of sections and subsections should be ordered. Currently, some subsections are not properly numbered.
- In section 2 two-level numbering should be entered.
- There are duplicate sections 3 („3. Materials and Methods” – line 113 and „3. Results” – line 244).
- What does "the transportation calculating formula" mean (line 135)?
- I don't understand on what basis the values of WRailway and WHighway have been taken as simple weighted averages? This should be properly proven. The question is whether it is not too far simplification?
- In my opinion, the explained variable (the index of ULUE undestood as „the ratio of output values of the secondary and tertiary industries to each provincial land area”) should also be defined formally by a formula.
- There is a mismatch in the notation, eg „xi, xj respectively represent the ULUE of region i and region j” (line 177) and „Xit represents the explanatory variable matrix, including transportation” (line 202-203). This may cause some confusion in the proper understanding of the text.
- No explanation of the symbols used in the formulas (9), (10) and (11). This should be completed.
- In my opinion, the phrase "TotalTrans represents the level of transportation infrastructure" (line 236) is not enough. The method of determining the level of transport infrastructure should be explained in more detail.
- The discussion section seems to be too short. I suggest extending the comments contained in the previous section.
- There is no description of the study area. In view of large deviations of the values of the variables (especially the explained variable) in individual provinces of China, the text of the article should be supplemented with maps showing the levels of these values, marked with different colors. In addition, detailed values of all variables (including control variables) should be included in the appendix.
Round 2
Reviewer 2 Report
The author has successfully addressed all my questions. Manuscript don't require further revision. I recommend accepting the manuscript in the present form.
Author Response
Dear Reviewer:
Thanks for your constructive remarks and useful suggestions, which have significantly raised the quality of the manuscript and have enabled us to improve the manuscript.
Thanks again!
Xinhai Lu , Mengcheng Wang , Yifeng Tang
Reviewer 3 Report
- I have read the paper after revision thoroughly. The quality has significantly improved and the majority of remarks were implemented.
- The only remark that left for the discussion is the lack of results presenting differences between all 30 regions being researched. In my opinion, it would be one of the most valuable results of this research. Therefore, I put my question&remark again: "Are there several similar "clusters"/groups of the regions? Are all regions similar in the case of the results obtained?" Figure 3 shows substantial differences between regions in case of ULUE in the period 2003-2018.
- For this specific paper, I recommend merging chapters 4 (Results) and 5 (Discussion) into one with the exemplary title "Results and discussion". Part of interesting and valuable discussion is now located in Chapter 4. Chapter 5 could become subchapter 4.5. But I leave it to the own decision of Authors.
Reviewer 4 Report
Dear Authors,
Thank you kindly for taking into account my comments in the revised version of your manuscript. I see, that the text has been significantly changed, which—in my opinion—has improved its quality.
However, I have noticed some minor shortcomings that should be corrected:
- The measure „Transi” (formula (2) – line 208) represents the level of transportation infrastructure of each province of China. In my opinion, in this formula the variables „Railway” and „Highway” should also be indexed by „i” (i.e. „Railwayi” and „Highwayi” respectively), because they refer to each individual province.
- The formulas should be completed with the units in which the resulting numerical values are expressed (especially in the case of formulas (1) and (2)).
- The text on lines 221-223 is in italics. Was this the intention of the Authors or an editorial mistake only?
- In reference item number 36 (line 574), the name of one of the authors is „Bragan?a”. Please check if this is correct.
Thank you!
